# PLANNING AT INFERENCE: MCTS TEST-TIME SCALING FOR LONG VIDEO GENERATION

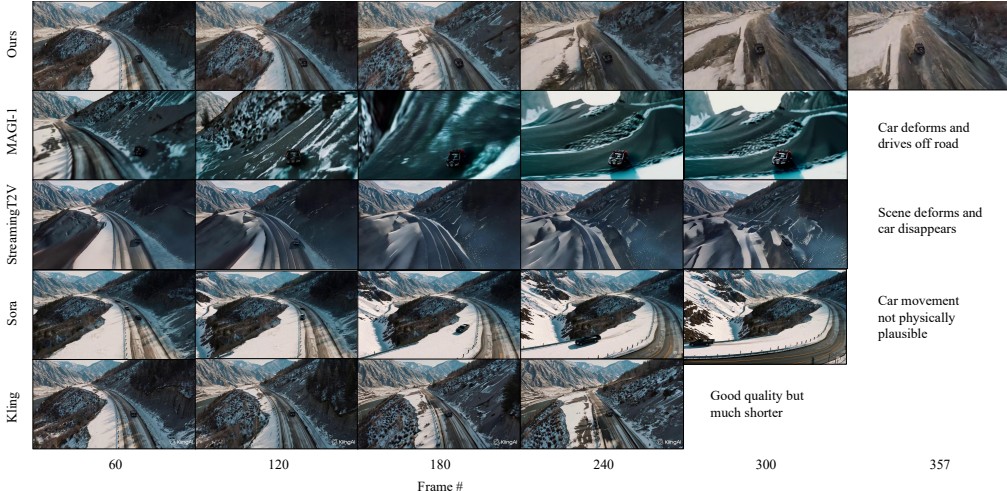

Figure 1: Long videos generated with our method vs other long video generation frameworks

## ABSTRACT

Generating long videos with consistent content and visual quality remains a major challenge, as existing one-shot and chunked methods often suffer from semantic drift and compounding artifacts. We explore Test-Time Scaling (TTS) as a framework for long video generation, formulating the task as a sequential decision-making problem. Our approach uses Monte Carlo Tree Search (MCTS) to evaluate multiple continuations with look-ahead rollouts and backpropagated rewards, and we introduce a Multi-Tree MCTS variant that improves exploration in continuous generation spaces. The method is modular and can be applied to existing backbones without retraining. Experiments on Cosmos-Predict2 and other models show consistent improvements in object permanence, temporal coherence, and text-video alignment over Best-of-N, Greedy, and Beam search. Furthermore, our method produces high-quality videos exceeding 20 seconds, surpassing the output of leading models like Sora and Kling by 18% and 47% respectively, all while maintaining comparable visual fidelity. Although the results are limited by the quality of current generators and verifiers, our study highlights both the promise of search-based TTS and the limitations of today's video generation and evaluation models.

## 1 INTRODUCTION

The field of video generation has advanced rapidly with the development of autoregressive (Arnab et al., 2021; Van Den Oord et al., 2017) and diffusion based approaches(Ho et al., 2022; Peebles & Xie, 2023; Rombach et al., 2022). These methods can now produce short clips with impressive fidelity and realism (Agarwal et al., 2025; ai et al., 2025; Zheng et al., 2024; Yang et al., 2024). However, extending this success to long-form video remains a fundamental challenge. As generation length increases, models face distributional shift, compounding error, and a breakdown of long-range temporal consistency (Wang et al., 2023), (Dalal et al., 2025). Consequently, videos often degrade rapidly, with semantic drift, motion decay, and visual artifacts accumulating over time.

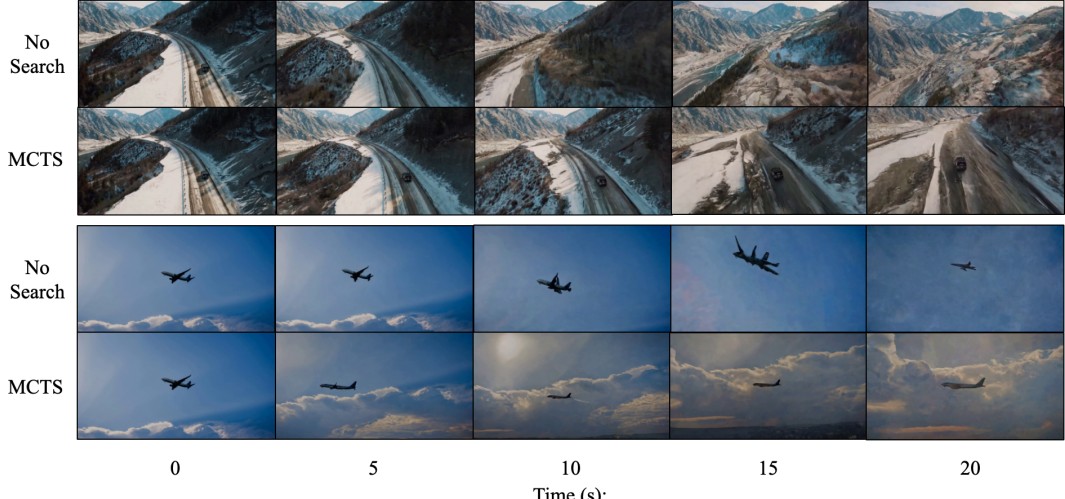

Figure 2: Videos generated with search display improved aesthetic qualtity and temporal consistency relative to the autoregressive baseline

Current approaches to long video generation fall into two categories. The first generates the entire sequence in one shot using autoregressive or diffusion backbones. While this can, in principle, capture global context, it is computationally prohibitive, even with highly compressed tokenizers. A more practical alternative is to generate videos autoregressively in chunks, extending the sequence segment by segment. This approach mirrors language modeling, where the distribution over a video $X = (x_0, \ldots, x_n)$ is factorized as

$$p(X) = p(x_0) \prod_{i=1}^{n} p(x_i | x_{i-1}, c) \tag{1}$$

with each chunk $x_i$ conditioned on its predecessor $x_{i-1}$ and optional context $c$ (e.g., a text prompt). This strategy is widely used, as it can be layered on top of any Image-to-Video (I2V) or Video-to-Video (V2V) model.

However, this autoregressive approach suffers from a critical flaw: **a lack of decision-making**. The autoregressive approach generates only a single continuation - disregarding possibilities that would be aesthetically and temporally superior. We argue that long video generation should but treated as a **sequential decision-making problem**. We propose modeling generation as a **Markov Decision Process** and utilize **Monte Carlo Tree Search** for optimizing the generation trajectory and producing high quality long videos.

Existing work has applied Test Time Scaling (TTS) at the frame or chunk level for generating higher quality videos, however, there is little work addressing the problem of long video generation with TTS. The standard search algorithms for TTS are best-of-N, greedy, and beam, however, we propose to use Monte Carlo Tree Search (MCTS) for generating flexible search trees and exposing high quality long videos. Furthermore, we introduce a novel Multi-Tree MCTS approach which we demonstrate has better properties for continuous search spaces.

Our contributions are threefold:

- We reformulate the task of long-video generation as a sequential decision-making process, allowing us to apply test-time scaling to significantly improve aesthetic quality and temporal coherence. Our method generates videos which are 18% longer and more consistent than Sora, and 47% longer than Kling while maintaining comparable fidelity.

- We propose a novel MCTS methodology for long video generation and introduce a novel multi-tree variant specifically designed for the optimization landscape of video generation.

- We address the limitations of video generation and visual quality assessment models, highlighting core areas that require improvement.

|  | MCTS (Search) | Autoregressive (No Search) | % Improvement |
|---|---|---|---|
| Cosmos-Predict2 | 8.405 | 7.872 | 6.77 |
| CogVideoX-5B | 8.276 | 7.514 | 10.14 |
| PyramidFlow (SD3) | 7.664 | 7.197 | 6.49 |
| SVD | 8.148 | 7.849 | 3.81 |
| Wan 2.2 | 8.486 | 8.124 | 4.46 |

Table 1: Applying MCTS-based search consistently improves video quality scores over the autoregressive baseline across a diverse set of backbone models. We observe a 10% improvement for the CogVideoX-5B model, illustrating the benefit of search based test time scaling.

## 2 RELATED WORK

### 2.1 TEST-TIME SCALING AND SEARCH-BASED INFERENCE

Test-time scaling (TTS) increases a model's computational budget at inference to boost performance without additional training. A key direction within TTS is the use of search algorithms to more effectively explore the model's output space, a strategy particularly prevalent in language modeling. For instance, **Tree-of-Thought (ToT)** Yao et al. (2023) and similar methods (Besta et al., 2024), (Zhang et al., 2024), (Hao et al., 2023) extend beyond single-chain reasoning by constructing and evaluating multiple reasoning paths.

This principle has been widely instantiated through Monte Carlo Tree Search (MCTS) across domains. For example, AlphaGo Zero leveraged MCTS to refine its policy network during self-play, illustrating the power of search in complex decision spaces Silver et al. (2018). In reasoning tasks, methods such as ReST-MCTS* Zhang et al. (2024) and the approach of Hao et al. (2023) employ MCTS to navigate trees of reasoning steps, guided by reward models that evaluate intermediate states. Similarly, DeepSeek-Prover-V1.5 applies MCTS to automated theorem proving by selecting and expanding proof tactics Xin et al. (2024).

Search is also gaining traction in generative vision. GenArtist frames image synthesis as a planning problem, constructing a tree of generation, editing, and verification steps Zhenyu et al. (2024). For diffusion models, T-SCEND explores alternative denoising trajectories with MCTS, incorporating future information into current updates Zhang et al. (2025). Other work applies beam search to probe multiple latent trajectories during denoising, retaining candidates scored highest by verifier models Liu et al. (2025); Cong et al. (2025a). Building on these principles, we employ search-based TTS for the sequential decision-making problem of long video generation.

An additional motivation for using MCTS comes from two previous works applying MCTS for learning search space partitions for multi-objective and black box optimizations Zhao et al. (2022), Wang et al. (2022). Both settings share key structural similarities with long video generation: the search space is vast, evaluations are costly, and the underlying objective is often non-convex and poorly behaved.

### 2.2 LONG-VIDEO GENERATION

Extending video generation beyond short clips remains a critical challenge, particularly in maintaining temporal consistency and avoiding semantic drift. Many current methods adopt an autoregressive, chunk-based strategy. For instance, MAGI-1 generates overlapping segments conditioned on prior outputs and employs pipelining for efficiency ai et al. (2025). Similarly, Gen-L-Video interpolates text embeddings across overlapping chunks to promote smooth semantic transitions Wang et al. (2023).

To improve long-range coherence, models often incorporate memory or adaptation mechanisms. Streaming-T2V uses a sliding attention window over previous frames and an appearance preservation module that conditions on the first frame Henschel et al. (2025). Test-Time-Training Dalal et al. (2025), Sun et al. (2025) introduces RNNs with expressive hidden states whose parameters are updated at each time step using the input context, enabling the model to directly integrate the current context into its weights and output distribution. Alternatively, SlowFast-VGen Hong et al.

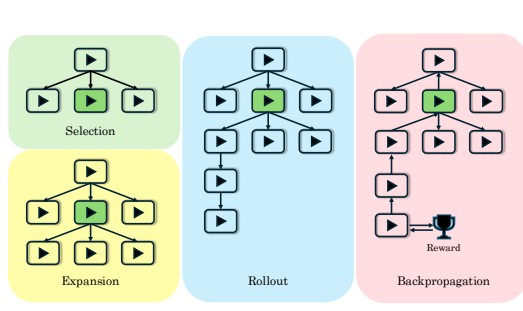

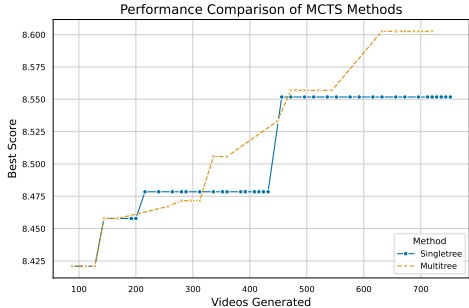

(a) Four phases of Monte Carlo Tree Search

(b) Multi-Tree outperforms Single-Tree MCTS

Figure 3: Test-time scaling with Monte Carlo Tree Search

(2024) implements a form of test-time training by using Low-Rank Adapters (LoRA) as a long-term memory bank, which is updated after generating each new chunk.

While these methods improve temporal consistency, they remain fundamentally one-step-ahead generation processes. Our approach is complementary: instead of modifying the backbone with memory or test-time training, we introduce a search-based decision process that can be applied on top of existing models to produce coherent long videos.

## 3 TEST TIME SCALING

### 3.1 AUTOREGRESSIVE VS SEARCH-BASED GENERATION

A long video $X$ can be represented as a sequence of discrete video segments, or chunks, $X = (x_0, x_1, \ldots, x_T)$. A standard approach is to generate this sequence autoregressively, where each new chunk $x_t$ is generated by a backbone model $\mathcal{G}$ conditioned on the previously generated chunk $x_{t-1}$ and global context $c$ (i.e. a text prompt).

To overcome the limitations of autoregressive decoding, we can employ search algorithms to explore a wider space of possible video sequences. This requires a mechanism to evaluate the quality of partial or complete sequences. We define two types of reward models for this purpose:

- **Process Reward Model (PRM)**, $V_p(x_t)$, evaluates the quality of an *individual* video chunk $x_t$. The PRM should assess properties such as visual and motion quality.
- **Outcome Reward Model (ORM)**, $V_o(X)$, evaluates the quality of a *complete* video sequence $X$. The ORM should assess properties such as temporal consistency and overall narrative coherence.

### 3.2 BASELINE SEARCH METHODS

The simplest search strategy is **Best-of-N** which generates $N$ independent videos, assesses them using the ORM, and selects the highest scoring video. Best-of-N does not utilize decision making at the chunk level, and does not adequately address the issue of compounding error.

The simplest chunk-wise search method is **Greedy Search**. At each step $t$, we generate a set of $M$ plausible continuations and select the chunk with the highest reward as determined by the process reward model $V_p$. While computationally efficient, greedy search suffers from myopia: locally optimal actions are chosen without considering their downstream impact on the overall video. This often leads to semantic drift and logical inconsistencies over long sequences.

A direct improvement over Greedy Search is **Beam Search**. At each step $t$, we maintain a "beam" of $K$ candidate sequences. For each of the $K$ sequences, we generate $M$ possible next chunks, producing $K \times M$ new candidates. The PRM is used to score these new candidates, and we retain the top $K$. While more robust than greedy search, beam search prematurely prunes superior paths due to its greedy nature.

## 3.3 Long-Video Generation with MCTS

To enable flexible and farsighted video generation, we introduce a search framework based on Monte Carlo Tree Search (MCTS). In our setting, the search tree consists of nodes representing video chunks and edges representing the generation of a child chunk from a parent. Critically, our approach modifies the standard MCTS objective: instead of identifying the most promising initial action, our goal is to discover the highest-quality complete video sequence found during the search. As illustrated in Fig. 3a, each MCTS iteration consists of four phases:

**1. Selection:** Starting from the root node (initial video chunk $x_0$), we traverse the tree by recursively selecting the child with the highest Upper Confidence Bound (UCB) score. This score balances exploitation - choosing nodes with high known rewards - and exploration - choosing less-frequently visited nodes:

$$\text{UCB}(v_i) = Q(v_i) + C\sqrt{\frac{\ln N_p}{N_{v_i}}} \qquad (2)$$

$Q(v_i)$ is the average reward of the child node $v_i$, $N_p$ is the visit count of the parent node, $N_{v_i}$ is the visit count of the child node, and $C$ is an exploration constant. This process continues until a leaf node $v_L$ is reached (a node that has not yet been expanded).

**2. Expansion:** If the leaf node $v_L$ is not a terminal state (the video is not yet complete), we expand it by using the backbone generator $\mathcal{G}$ to create a fixed number of children (continuations). We can optionally use a tailored prompt for each child to encourage diversity, a form of reasoning-based scaling.

**3. Simulation (Rollout):** The simulation phase generates a complete video sequence, $X_{rollout}$ by continuing from the newly expanded node. The quality of this complete video is then estimated by the ORM, yielding a reward $R = V_o(X_{rollout})$. This reward estimates the long-term value of being in that state, and we keep track of the best rollout seen so far as determined by the ORM.

**4. Backpropagation:** The rollout reward $R$ is backpropagated up the tree from the rollout leaf to the root. For each node v along this traversal path, its visit count $N_v$ is incremented, and its average reward $Q(v)$ is updated to incorporate the information gained from the simulation.

After running a fixed number of MCTS iterations, the final video sequence is generated by concatenating all the chunks in the rollout with the highest reward. To improve the efficacy of MCTS for video generation, we introduce modifications to the standard paradigm:

- **Parallel Tree Expansions:** Inspired by AlphaGo-Zero Silver et al. (2018), we perform tree expansions in parallel. This strategy rapidly expands the search frontier, allowing for exploring diverse trajectories in parallel while minimizing investment in suboptimal regions of the search space.
- **Hybrid Beam-MCTS Initialization:** We initialize the tree using beam search for the first few steps, a technique partially adapted from T-SCEND Zhang et al. (2025). This ensures a broad and diverse initial search space, enabling subsequent MCTS iterations to focus on refining the most promising sub-trees.
- **Greedy Rollouts:** We employ greedy rollouts during the simulation phase to efficiently estimate the maximum achievable reward from any node. This approach improves the estimation accuracy of the maximum achievable reward, and improves final video quality without increasing wall clock time, as the rollouts can be fully parallelized at each depth of the search tree.

## 3.4 Multi-Tree MCTS

A key limitation of standard MCTS in continuous state spaces is that, with a fixed branching factor, exploration remains confined to the subspace defined by the root's initial children. This can trap the search in local optima while leaving other promising regions unexplored, effectively capping the achievable video quality. Continuation-action variants expand the search more broadly but are prohibitively expensive in the context of video generation.

|  | Avg. Score | # of Video Chunks | GPU Hours |
|---|---|---|---|
| Autoregressive | 7.872 | 4 | 0.5 |
| Best-of-N (N=2) | 7.961 | 8 | 1 |
| Greedy | 8.286 | 32 | 4 |
| Beam (K=2) | 8.325 | 56 | 7 |
| MCTS | **8.405** | 160 | 20 |

Table 2: MCTS generates higher quality videos than the baseline methods. This can be attributed to a more effective use of additional compute. However, consistent with inference time scaling laws, the returns diminish with additional compute.

To address this, we propose Multi-Tree MCTS, a variant designed to systematically broaden the search space. Instead of exhaustively optimizing a single tree, our method initiates a new search tree once the current tree exceeds a predefined size. This strategy enforces exploration at a higher level, allowing the algorithm to investigate multiple, orthogonal regions of the search space. By doing so, Multi-Tree MCTS is less susceptible to suboptimal initializations and is more likely to discover subspaces that contain higher-quality optima.

## 4 EXPERIMENTS

### 4.1 EXPERIMENTAL SETUP

**Generation backbone:** We use Cosmos-Predict2 Agarwal et al. (2025) as our video generation backbone. The model natively produces 5s clips at 16 FPS and cannot generate longer sequences in a single pass. Using search, we extend this to 20s videos—four times its base length.

**Reward model:** Our process reward model (PRM) combines three components: VideoScore, CLIP-based scores, and the LAION perceptual model. VideoScore He et al. (2024) is a state-of-the-art no-reference VQA model shown to correlate with VBench Huang et al. (2024). CLIP embeddings capture text–video alignment and frame consistency, while the LAION model assesses perceptual quality, following SWIFT Cong et al. (2025b), which demonstrated their correlation with VBench and human preference. Because no existing VQA model reliably handles long-form video, we define the outcome reward model (ORM) as the sum of PRM scores across all chunks.

### 4.2 SCALING INFERENCE BUDGET IMPROVES GENERATION QUALITY

Our central assumption is that increasing inference-time compute improves video generation quality. To test this, we begin with a simple Best-of-N (BoN) strategy: generate N candidate videos and select the one with the highest reward score.

As shown in Fig. 5a, video quality increases monotonically with N, though with diminishing returns. This reflects both the high variance of generated outputs and the non-convexity of the underlying search space: exploring more trajectories exposes additional local optima. These results empirically support our hypothesis that test-time scaling is beneficial even under a naive strategy, motivating the use of more powerful search methods such as Monte Carlo Tree Search.

### 4.3 COMPARISON OF SEARCH METHODS

We evaluate our MCTS framework against a hierarchy of baselines: (1) No Search, (2) Best-of-N, (3) Greedy Search, and (4) Beam Search, which represent increasingly complex search strategies. Average scores are shown in Fig. 5b, with computational costs in Table 2.

**Baseline performance:** Even simple strategies outperform the autoregressive baseline. Best-of-N leverages variance across trajectories: sampling multiple candidates increases the chance of finding a higher-quality video. Greedy and Beam Search extend this to chunk-level decision-making, with Beam outperforming Greedy by maintaining K parallel trajectories. This diversity is important because the generation landscape is non-convex: locally optimal chunks may not yield globally

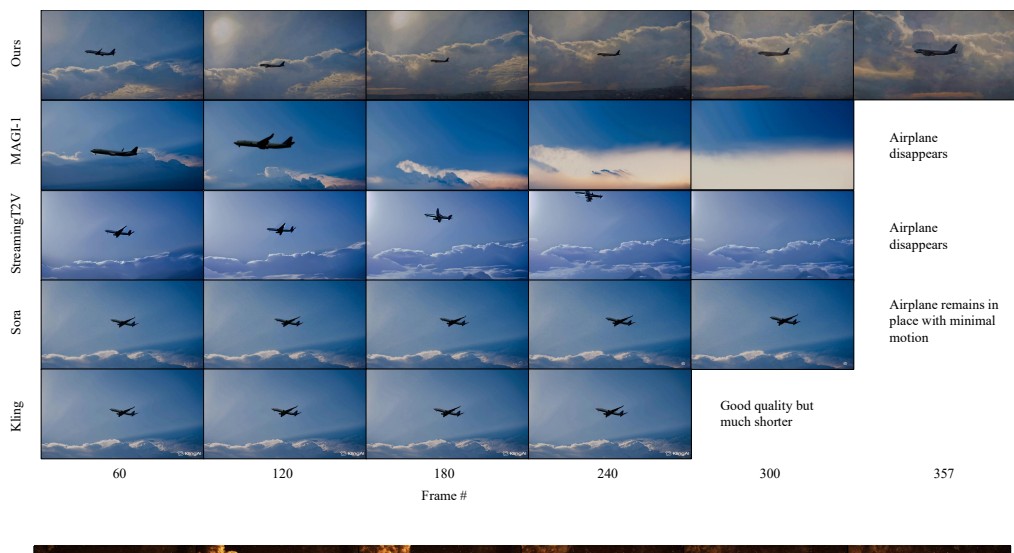

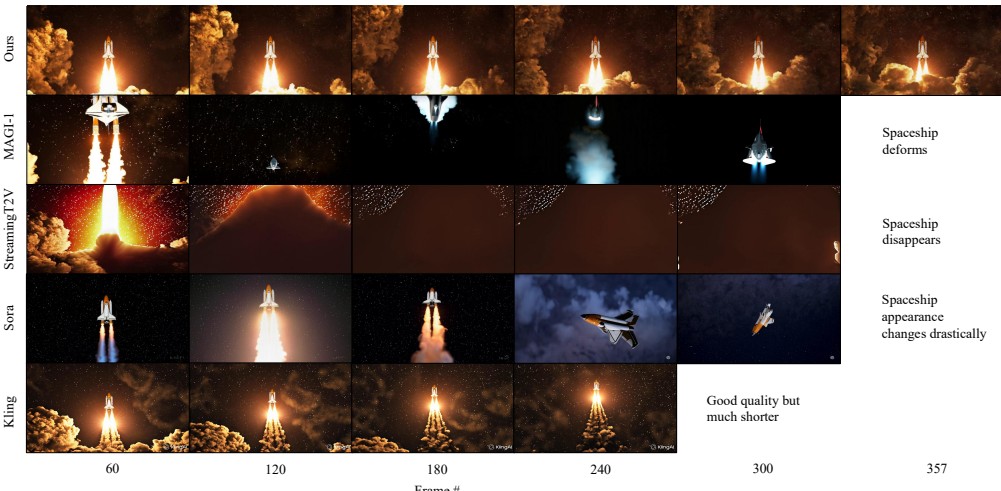

Figure 4: Our search method vs other long-video generation frameworks

coherent videos. By retaining multiple candidate paths, Beam Search better explores local optima, producing higher average quality.

**Superiority of MCTS Framework:** Following similar trends in black box optimization (Zhao et al., 2022), (Wang et al., 2022), our MCTS framework outperforms all baselines for long video generation. This superior performance stems from its unique synthesis of exploration and exploitation, which overcomes the limitations of nearsighted methods such as Beam and Greedy Search (Yang et al., 2021), (Wang et al., 2021).

1. **Effective Exploration**: Unlike beam search, which discards trajectories not contained in the set of beams, MCTS supports backtracking. This allows the search to escape local optima by re-investigating previously less-promising paths that lead to superior long-term outcomes. This ability to backtrack and explore is critical for navigating the complex video generation landscape.

2. **Informed Exploitation**: The backpropagation of rollout rewards allows MCTS to incorporate information about the future into present decisions. A path that appears immediately suboptimal may be selected if it leads to high-reward futures. This forward-looking decision-making process enables MCTS to holistically evaluate continuations and identify optimal trajectories that other methods cannot.

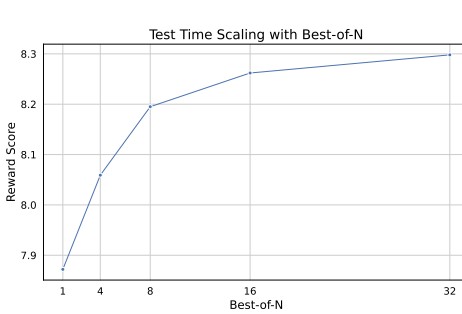

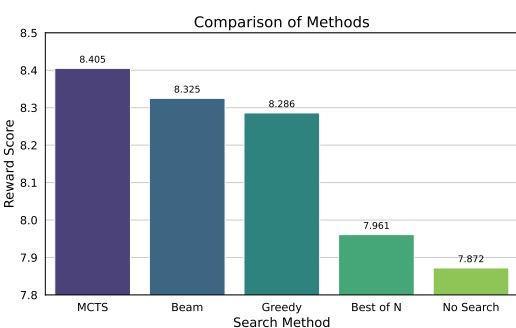

(a) Test time scaling improves the quality of generated videos, albeit with diminishing returns

(b) MCTS generates higher quality videos than the baseline search methods

Figure 5: Test-time scaling with Monte Carlo Tree Search

Finally, the superior performance of our framework is amplified by several modifications designed to maximize search efficiency. Firstly, parallel tree expansions rapidly broaden the search tree, generating multiple rollouts in parallel. Secondly, a beam search initialization phase establishes a strong and diverse set of initial candidate trajectories, allowing subsequent MCTS iterations to focus on refining the most promising sub-trees. Finally, the use of greedy rollouts leads to the discovery of superior final videos. Each modification contributes to a more effective allocation of our computational budget.

### 4.4 MCTS vs Multi-Tree MCTS

As introduced in Section 3.4, our Multi-Tree MCTS algorithm is designed to enhance search space exploration. It addresses the fundamental trade-off between standard MCTS, which risks under-exploration by optimizing within a narrow area, and continuous action variants, whose broad exploration strategy is often computationally prohibitive for video generation. Our approach offers a practical solution by generating multiple independent search trees, encouraging exploration of diverse regions and making the search less susceptible to poor initializations.

As illustrated in Fig. 3b, multi-tree MCTS achieves a better exploration–exploitation balance than its single-tree counterpart. For the same computational budget, maintaining multiple search trees yields a more diverse set of high-quality optima than exhaustively expanding a single tree.

As shown in our results, multi-tree MCTS not only attains higher quality scores but also sustains improvements beyond the plateau of single-tree search. This makes it a more robust strategy in high-compute regimes. An open question is how best to allocate a fixed budget between the number of trees and the depth of exploration within each tree.

### 4.5 Qualitative Results

Our search-based approach produces videos with improved object permanence, temporal consistency, and text–video alignment. As shown in Fig. 2, videos generated with our framework degrade more slowly over time, maintaining higher spatial and temporal coherence than autoregressive baselines. For example, in the first setting, the car remains visible throughout the MCTS-generated video, whereas it disappears without search. This illustrates how MCTS-based TTS actively guides the generation process toward higher-quality outputs.

Compared to other long-video generation frameworks, our method achieves superior quality and longer durations. In Fig. 1, the MCTS-generated video preserves the car's presence and motion, while it vanishes in outputs from MAGI-1, StreamingT2V, and Sora. Similar improvements are observed in Fig. 4. The exception is Kling, which produces competitive quality; however, our method generates videos that are 47% longer while maintaining comparable fidelity.

|  | NIQE↓ | BRISQUE↓ | T.C.↑ | S.C.↑ | A.Q↑ | M.S.↑ |
|---|---|---|---|---|---|---|
| Ours | 5.938 | 40.526 | 7.67 | 8.33 | 8 | 8.33 |
| MAGI | 8.420 | 66.222 | 1.67 | 1 | 2.33 | 2.33 |
| StreamingT2V | 7.328 | 58.423 | 2.67 | 1.67 | 3 | 3.33 |
| Sora | 6.904 | 31.724 | 6.67 | 7 | 8 | 6 |
| Kling | 5.638 | 38.166 | 9.33 | 10 | 9.33 | 7 |

Table 3: **Quantitative comparison of long-video generation frameworks.** The models are evaluated using no-reference perceptual metrics (NIQE, BRISQUE; lower is better) and scores from Gemini 2.5 Pro for Temporal Consistency (T.C.), Subject Consistency (S.C.), Aesthetic Quality (A.Q.), and Motion Smoothness (M.S.) (higher is better).

### 4.6 COMPARISON OF GENERATED VIDEOS

To validate our approach, we benchmarked it against leading frameworks using perceptual metrics (NIQE, BRISQUE) and qualitative scores generated by Gemini 2.5 Pro Comanici et al. (2025) Table 3. Our method demonstrates a clear advantage, outperforming MAGI and StreamingT2V across all evaluated criteria. Notably, it also surpasses Sora in four out of six metrics and achieves the highest score for motion smoothness (8.33) among all baselines. This superior performance in motion highlights our search-based approach's effectiveness in generating consistent and natural video continuations.

## 5 SUGGESTIONS FOR FUTURE WORKS

Our method searches for high-quality samples within the generator's learned distribution, as judged by a reward model. Its performance is therefore upper-bounded by the capabilities of both the video generation backbone and the verifier. Consequently, if the backbone cannot produce coherent video segments, or if the reward fails to accurately assess them, search cannot fully compensate. Thus, these components need to be improved to push the upper bound of long video generation.

**Backbone limitations.** Current models degrade over long horizons due to distributional shift: trained on extending real images and videos, they struggle when conditioned on their own imperfect outputs. This leads to motion decay, frozen frames, and increasing artifacts. Furthermore, we found that conditioning on frames from previous chunks can sometimes revive motion but does so inconsistently, and chunk-wise prompting does not resolve this issue. Search can slow this degradation by selecting better continuations, however, the backbone remains the primary bottleneck.

**Reward model limitations.** Existing verifiers (e.g., CLIP-based metrics, VideoScore) misalign with human preferences, especially for long-form content. They fail to capture temporal consistency and text-video alignment, and their limited frame windows (24–48 frames) reduce effectiveness for longer clips. As a result, search may optimize for signals that do not reflect perceptual quality. Progress requires more reliable long-video VQA models that assess coherence across full sequences; such models would not only improve our MCTS framework but also provide stronger evaluation benchmarks for the field.

## 6 CONCLUSION

We cast long video generation as a sequential decision-making problem solvable with test-time scaling. Our approach uses Monte Carlo Tree Search to navigate the vast generation space, outperforming standard search baselines by balancing exploration and exploitation. We also introduce Multi-Tree MCTS, a variant with superior scaling properties that is better suited to efficiently navigate continuous state spaces. While our method is effective, it also highlights critical areas for future work: developing more efficient search algorithms, better reward models, and stronger video generation backbones. Ultimately, framing generation as a planning problem is a powerful and promising paradigm for creating coherent, long-form content.

## 7 ETHICS STATEMENT

The research presented in this paper aims to address the challenges of long video generation. We acknowledge that generative video models, like the ones used in our study, have the potential for misuse, such as the creation of convincing misinformation or harmful synthetic video. However, our work focuses on improving the coherence and quality of generated content through a search-based inference-time technique, rather than developing new foundational models. The method itself is neutral and does not inherently increase the risks associated with the underlying generative backbones. All experiments were conducted using publicly cited models. We are committed to responsible AI and believe that advancing the technical understanding of these systems is a crucial step toward developing effective safeguards and detection methods. We encourage the community to continue research into the ethical deployment of these generative models.

## 8 REPRODUCIBILITY STATEMENT

To ensure the reproducibility of our results, we will make our code, experimental setup, and evaluation materials publicly available upon publication.

Code: The implementation of our Monte Carlo Tree Search (MCTS) framework, the Multi-Tree MCTS variant, and all baseline search algorithms (Best-of-N, Greedy, and Beam Search) will be released in a GitHub repository. The repository will also include scripts to replicate the main experiments and generate figures from the paper.

Models and Environment: Our primary generation backbone is Cosmos-Predict2, a publicly available model. The process reward model (PRM) is a composite of publicly available components: VideoScore , CLIP-based scores, and the LAION perceptual model. All experiments were conducted using NVIDIA H100 GPUs.

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

## A    APPENDIX: ALGORITHM PSEUDOCODE

The appendix contains pseudocode for Greedy Search, Beam Search, and Multi-Tree MCTS to complement the high-level descriptions in the main text.

---

**Algorithm 1** Greedy Search for Long-Video Generation

---

**Require:** Video generator $\mathcal{G}$, Process Reward Model $V_p$, initial chunk $x_0$, conditioning prompt $c$, sequence length $T$, number of continuations $M$.

1: Initialize video sequence $\mathcal{X} \leftarrow [x_0]$
2: **for** $t = 1$ to $T - 1$ **do**
3:     $x_{prev} \leftarrow \mathcal{X}[t-1]$
4:     Initialize candidate set $\mathcal{C} \leftarrow \{\}$
5:     **for** $j = 1$ to $M$ **do**
6:         $x_{candidate} \leftarrow \mathcal{G}(x_{prev}, c)$
7:         $r_{candidate} \leftarrow V_p(x_{candidate})$
8:         Add $(x_{candidate}, r_{candidate})$ to $\mathcal{C}$
9:     **end for**
10:    $\hat{x}_t \leftarrow \arg\max_{(x,r)\in\mathcal{C}} r$ {Select chunk with the highest reward}
11:    Append $\hat{x}_t$ to $\mathcal{X}$
12: **end for**
13: **return** $\mathcal{X}$

---

**Algorithm 2** Beam Search Decoding for Long-Video Generation

---

**Require:** Video generator $\mathcal{G}$, Process Reward Model $V_p$, initial chunk $x_0$, sequence length $T$, beam width $K$, num continuations $M$.

1: Initialize beams $\mathcal{B} \leftarrow \{([x_0], 0)\}$ {A set of (sequence, score) tuples}
2: **for** $t = 1$ to $T - 1$ **do**
3:     Initialize all candidates $\mathcal{C}_{all} \leftarrow \{\}$
4:     **for** each $(\mathcal{X}_{seq}, s_{seq})$ in $\mathcal{B}$ **do**
5:         $x_{prev} \leftarrow \mathcal{X}_{seq}.\text{last}()$ {Get the last chunk of the current beam}
6:         **for** $j = 1$ to $M$ **do**
7:             $x_{candidate} \leftarrow \mathcal{G}(x_{prev})$
8:             $\mathcal{X}_{new} \leftarrow \mathcal{X}_{seq} + [x_{candidate}]$ {Create a new candidate sequence}
9:             $s_{new} \leftarrow s_{seq} + V_p(x_{candidate})$ {Update the cumulative score}
10:            Add $(\mathcal{X}_{new}, s_{new})$ to $\mathcal{C}_{all}$
11:        **end for**
12:    **end for**
13:    $\mathcal{B} \leftarrow \text{Top-K}_{(\mathcal{X},s)\in\mathcal{C}_{all}} s$ {Select the top K candidates for the new beams}
14: **end for**
15: $(\mathcal{X}_{best}, s_{best}) \leftarrow \arg\max_{(\mathcal{X},s)\in\mathcal{B}} s$ {Select the best sequence from the final beams}
16: **return** $\mathcal{X}_{best}$

---

---

**Algorithm 3** Our MCTS

---

**Require:** Video generator $\mathcal{G}$, initial chunk $x_0$, iterations $N$, PRM $V_p$, ORM $V_o$, Time $T$, Beam time $T'$.
1: Initialize tree $\mathcal{T}$ with root node $v_{root}$ representing $x_0$.
2: Run Beam Search on $v_{root}$ till time $T'$
3: **for** $i = 1$ to $N$ **do**
4:     *// 1. Selection Phase*
5:     $v_{leaf} \leftarrow$ SelectLeafNode$(\mathcal{T}, v_{root})$
6:     *// 2. Expansion Phase*
7:     ExpandNode$(\mathcal{T}, v_{leaf}, \mathcal{G}, V_p)$
8:     *// 3. Simulation Phase*
9:     $v_{terminal},$R $\leftarrow$ SimulateRollout$(v_{leaf}, \mathcal{G}, V_o)$
10:    *// 4. Backpropagation Phase*
11:    Backpropagate$(\mathcal{T}, v_{terminal}, R)$
12: **end for**
13: *// Generate final video by selecting the path with the highest cumulative reward*
14: $\mathcal{X}_{final} \leftarrow$ GetBestPath$(\mathcal{T}, v_{root})$
15: **return** $\mathcal{X}_{final}$

---

**Algorithm 4** Multi-Tree MCTS

---

**Require:** Video generator $\mathcal{G}$, initial chunk $x_0$, iterations $N$, PRM $V_p$, ORM $V_o$, Time $T$, Beam time $T'$, Node budget $H$.
1: Initialize tree $\mathcal{T}$ with root node $v_{root}$ representing $x_0$.
2: Initialize $\mathcal{X}_{final}$ to $NULL$
3: Run Beam Search on $v_{root}$ till time $T'$
4: **for** $i = 1$ to $N$ **do**
5:     **if** NodeCount$(\mathcal{T}) > H$ **then**
6:        $\mathcal{X}_{current} \leftarrow$ GetBestPath$(\mathcal{T})$
7:        $s_{current} \leftarrow V_o(\mathcal{X}_{current})$
8:        **if** $s_{current} > V_o(\mathcal{X}_{final})$ **then**
9:           $\mathcal{X}_{final} \leftarrow \mathcal{X}_{current}$
10:       **end if**
11:       $\mathcal{T} \leftarrow$ InitializeTreeWithRoot$(x_0)$ {Start a new tree}
12:     **end if**
13:     *// 1. Selection Phase*
14:    $v_{path}, v_{leaf} \leftarrow$ SelectLeafNode$(\mathcal{T}, v_{root})$
15:    *// 2. Expansion Phase*
16:    ExpandNode$(\mathcal{T}, v_{leaf}, \mathcal{G}, V_p)$
17:    *// 3. Simulation Phase*
18:    $R \leftarrow$ SimulateRollout$(v_{leaf}, \mathcal{G}, V_o)$
19:    *// 4. Backpropagation Phase*
20:    Backpropagate$(\mathcal{T}, v_{path}, R)$
21: **end for**
22: *// Generate final video by selecting the path with the highest cumulative reward*
23: $\mathcal{X}_{final} \leftarrow$ GetBestPath$(\mathcal{T}, v_{root})$
24: **return** $\mathcal{X}_{final}$

---

## B  APPENDIX: ADDITIONAL RESULTS

| N (Best-of-N) | 1 | 4 | 8 | 16 | 32 |
|---|---|---|---|---|---|
| Avg. Score | 7.872 | 8.059 | 8.195 | 8.262 | 8.298 |
| *% Improvement* | – | | | | |

Table 4: Benefit of Scaling Inference Time Compute. The average score of generated videos increases monotonically with N, though with diminishing returns.

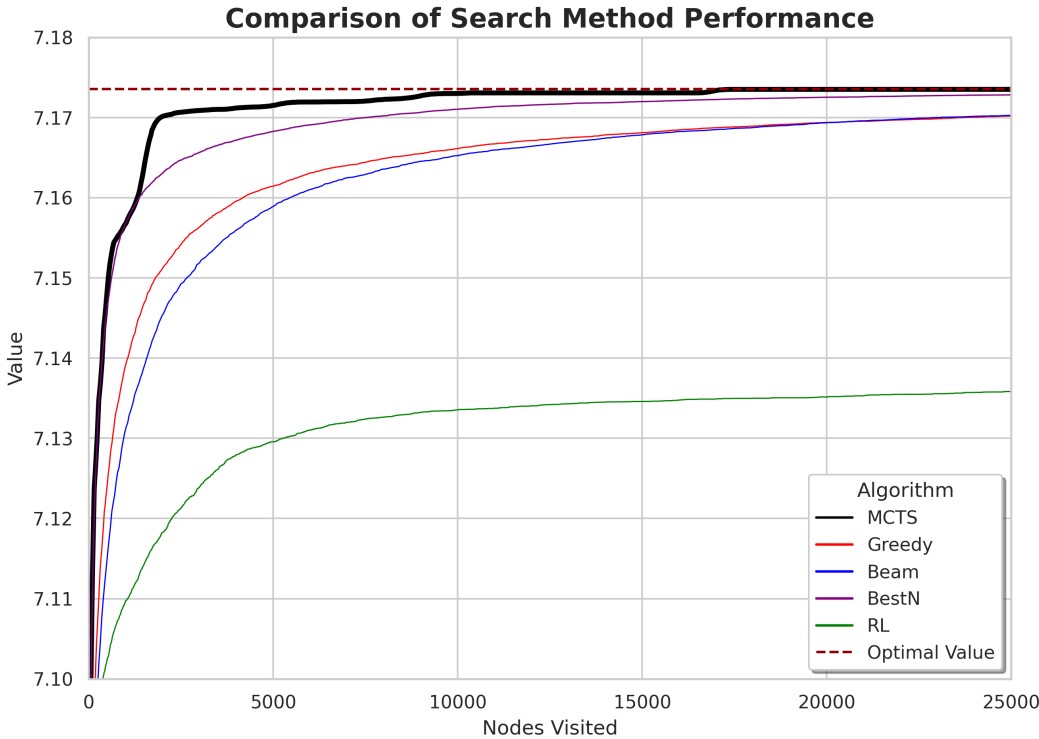

Figure 6: Search methods on full tree.

Our experiments confirm that Monte Carlo Tree Search (MCTS) is a remarkably efficient algorithm for navigating complex search spaces. We validated this by first constructing a fixed ground-truth tree of video sequences (depth 7, branching factor 3) with a known optimal solution. When benchmarked against other search paradigms on this tree, MCTS consistently located the global optimum with significantly greater speed and efficiency. This result underscores the algorithm's strength in effectively balancing exploration and exploitation to rapidly converge on the best possible outcome.

# C  APPENDIX: ADDITIONAL SUGGESTIONS FOR FUTURE WORKS

## C.1  REASONING

While our work focuses on search-based test-time scaling, an alternative is to modify the proposal distribution, as widely studied in LLMs. Reasoning chains, for instance, reshape the distribution over future tokens to improve output quality. In video generation, analogous techniques include chunk-wise tailored prompts or selecting conditioning frames, which aim to bias the model toward more coherent continuations. Mathematically, these methods reshape the conditional distribution to increase the likelihood of better continuations; intuitively, they supply information that helps the model preserve semantic and temporal consistency while improving visual quality.

In practice, however, we find that current I2V and V2V models struggle to leverage such conditioning. As shown in Table 5, VBench scores fail to improve when tailored prompts are applied. This suggests that these models often prioritize the image/video condition while under utilizing the textual context.

## C.2  SEARCH METHODS

We frame long video generation as a sequential decision-making problem and address it with an MCTS-based approach. While effective, our results highlight several directions for improving search efficiency.

One avenue is dynamic resource allocation, such as early stopping at the multi-tree level to reduce compute spent on low-performing subspaces. Such a mechanism would introduce exploration/exploitation at the multi-tree level, dynamically allocating compute based on the relative potential of a search space.

Another promising direction is to dynamically structure the search tree. Rather than a fixed branching factor, future methods could vary expansion by depth, wider early exploration with progressive narrowing, and diversity, adding children when existing candidates lack variability.

Finally, MCTS remains computationally expensive. More efficient variants, cheaper rollout approximations, or hybrid methods that integrate MCTS-style exploration into lighter strategies (e.g., beam or greedy search) may provide better cost–performance tradeoffs. All of these are exciting avenues for future research.

## C.3  REWARD MODELS AND VERIFIERS

The success of search-based methods relies on accurate reward signals to guide the exploration process. In the context of video, reward models are trained to predict human preferences and assess specific quality dimensions. For instance, **VideoScore** is a Vision-Language Model (VLM) trained to score videos on metrics like temporal consistency, visual quality, and text alignment He et al. (2024). Similarly, **VisionReward** learns to regress human preference scores by first training on binary quality-assessment questions and then fine-tuning with DPO Xu et al. (2025). These models serve as powerful, automated critics that can provide the necessary guidance for search algorithms. In our framework, we leverage such verifiers as process reward models (PRMs) to score the quality of intermediate video chunks, guiding our search toward globally coherent and high-quality final outputs.

Table 5: VBench comparison of videos generated with and without reasoning

|  | S.C. | B.C. | T.F. | M.S. | D.D. | A.Q. | Avg. |
|---|---|---|---|---|---|---|---|
| Reasoning | 0.9209 | 0.9478 | 0.9776 | 0.9876 | 0.8750 | 0.4847 | **0.8656** |
| No Reasoning | 0.9227 | 0.9499 | 0.9780 | 0.9878 | 0.8945 | 0.4837 | **0.8694** |

# D   APPENDIX: USAGE OF LLMS

The authors utilized Large Language Models (LLMs) to support the preparation of this manuscript. During the literature review phase, LLMs assisted in identifying and summarizing relevant prior work. In the revision stage, an LLM was used as a tool for proofreading, refining sentence structure, and enhancing the overall clarity of the text. The core conceptual contributions, experimental design, and interpretation of results are solely the work of the authors.

