# OpenReview forum: "Planning at Inference: MCTS Test-Time Scaling for Long Video Generation"
_ICLR.cc/2026/Conference — Submitted to ICLR 2026_

### Official Review · Reviewer_6AqY · 2025-10-30

**Soundness:** 3
**Presentation:** 2
**Contribution:** 2
**Rating:** 4
**Confidence:** 3

**Summary:**

This paper frames long video generation as sequential decision making and proposes test time search with Monte Carlo Tree Search to plan over chunked continuations, guided by a process reward model for local chunk quality and an outcome reward model that aggregates scores over the full sequence. The method is model agnostic, sits on top of existing backbones without retraining, and introduces a multi tree variant to widen exploration in continuous spaces. Across several generators, the approach improves temporal consistency and object permanence relative to autoregressive decoding, Best of N, greedy, and beam search, and reports longer, competitive quality videos when compared qualitatively and with automated metrics to recent long video systems. The paper provides algorithmic details, ablations on compute budget, and comparisons of single tree versus multi tree search, while also acknowledging dependencies on the underlying generator and verifier quality.

**Strengths:**

1. Clear formulation of long video generation as planning with Monte Carlo Tree Search, including a walk through of selection, expansion, rollout, and backpropagation plus an explicit UCB objective.

2. Multi tree search broadens exploration under a fixed branching factor and empirically outperforms single tree for the same budget.

3. Practical recipe that is plug in and does not require retraining, which increases utility for current systems constrained by backbone quality.

**Weaknesses:**

1. Heavy reliance on automated reward signals for both search guidance and evaluation, with outcome reward defined as a simple sum over chunks, risks overfitting to verifier idiosyncrasies rather than human preference on long horizon coherence. A controlled human study is missing.

2. The exploration constant, branching factor, rollout policy, and beam initialization depth can strongly affect MCTS behavior. Sensitivity analysis is not comprehensive.

**Questions:**

1. How sensitive are results to the weighting of VideoScore, CLIP alignment, and the LAION perceptual model in the process reward, and to the definition of the outcome reward as a sum rather than a learned temporal model

2. Under a fixed wall clock and identical hardware, how does the method compare to beam and greedy tuned for the same final runtime, including beam initialization time and rollout parallelism

---

> ### Author Response · Authors · 2025-12-02
> **Response to Reviewer 6AqY (Part 1/2)**
>
> ### Q1: Heavy reliance on automated reward signals for both search guidance and evaluation risks overfitting to verifier idiosyncrasies rather than human preference. ###
>
> We agree that the alignment between reward models and human preference is a critical area for improvement. However, MCTS specifically mitigates the risk of overfitting to the reward model via its Upper Confidence Bound (UCB) exploration term. This allows the model to explore a broader state space rather than exploiting specific reward artifacts, an advantage supported by (Wang et. al., Learning Search Space Partition for Black-box Optimization using Monte Carlo Tree Search; Zhao et. al., Multi-objective Optimization by Learning Space Partitions; Wang et. al., Neural Architecture Search using Deep Neural Networks and Monte Carlo Tree Search). Additionally, our qualitative analysis verifies that our model remains well-aligned with human preferences in practice: https://docs.google.com/presentation/d/1Agi0E049WxfRjtCEmpNyhvOzAWv_4g8yIqhGK6idSMA/edit?usp=sharing
>
> We agree that the ideal scenario involves integrating human preference feedback at every branching step; however, this remains computationally infeasible. The current Outcome Reward Model (ORM) formulation is necessary because **no established, meaningful reward models currently exist for assessing long video coherence** (20 or more seconds). Therefore, we use the local verifier model as the closest practical approximation.
>
> Our framework is future-proofed: as **external reward models improve, the performance of our MCTS framework will scale proportionally**. Our work should be assessed on the merits of its MCTS/TTS contribution, which shows promising results in scaling up generative video. We acknowledge that the creation of a superior visual reward model is an **orthogonal line of research**.
>
> &nbsp;
> ### Q2: The exploration constant, branching factor, rollout policy, and beam initialization depth can strongly affect MCTS behavior. Sensitivity analysis is not comprehensive. ###
>
> We agree that future sensitivity analysis will further solidify this paper and we will include this in the final version of the paper. We selected an exploration constant of $\sqrt{2}$, which represents approximately 15% of the measured reward range. The majority of the settings are consistent with optimal settings found in prior literature applying MCTS for black-box and Bayesian optimization (Wang et. al., Learning Search Space Partition for Black-box Optimization using Monte Carlo Tree Search; Zhao et. al., Multi-objective Optimization by Learning Space Partitions).
>
> Our choice of a branching factor of **8** was determined by hardware efficiency. This ensures full utilization of a compute node by running parallel inference across the **8 available GPUs** during the tree expansion phase.
> Preliminary experiments demonstrated that non-greedy (stochastic) rollouts were **computationally prohibitive** for high-quality video generation and **rapidly degraded the quality and coherence of individual video chunks**. The observed use of a greedy policy yielded a **substantially superior qualitative improvement** in generated video quality validating its adoption as the effective heuristic for this high-dimensional task.
>
> Beam initialization depth of **2** was chosen to be consistent with another paper (Zhang et. al., T-scend: Test-time scalable mcts-enhanced diffusion model) applying greedy search followed by MCTS for diffusion trajectories; making search more robust and efficient.

---

> ### Author Response · Authors · 2025-12-02
> **Response to Reviewer 6AqY (Part 2/2)**
>
> ### Q3: How sensitive are results to the weighting of VideoScore, CLIP alignment, and the LAION perceptual model in the process reward, and to the definition of the outcome reward as a sum rather than a learned temporal model. ###
>
> The choice of defining the outcome reward as a **simple sum over chunks** is pragmatic and defensible because it provides a viable objective given the current progress of video reward modeling. We confirm that a learned temporal model would theoretically be superior and is the optimal direction for future research, as it would significantly improve the performance of our MCTS approach We investigated nearly all existing open source models and found that none are adequate for scoring long videos - this was manually verified.
>
> We use VideoScore because it is the SOTA video reward model providing scores across multiple dimensions unlike the other two reward models. CLIP and LAION are used as they were used in prior work applying TTS for video generation (Zhao et. al., Can Test-Time Scaling Improve World Foundation Model), and were found to be highly beneficial. We don’t have an exact comparison of sensitivity, however, the quality of generated videos are more affected by the VideoScore model than the other two (as VideoScore captures more dimensions of video quality).
>
>
> &nbsp;
> ### Q4: Under a fixed wall clock and identical hardware, how does the method compare to beam and greedy tuned for the same final runtime, including beam initialization time and rollout parallelism. ###
> In Figure 6 in the appendix, we study the performance under fixed computational budgets and MCTS scales better than greedy, beam, and best of N.

---

### Official Review · Reviewer_UF3o · 2025-11-01

**Soundness:** 3
**Presentation:** 3
**Contribution:** 4
**Rating:** 6
**Confidence:** 3

**Summary:**

The paper proposes using MCTS for planning-based long video generation, which expands an important direction in the TTT field. Through this approach, the paper even achieves long video generation results that surpass closed-source SOTA models, demonstrating the potential of TTT in long video generation.

**Strengths:**

- The work has a certain degree of novelty and community value. The paper is the first to apply MCTS-based TTT to long video generation, showcasing the value of classical methods in the video domain.
- The experimental results are impressive. The proposed method enables Cosmos-Predict2 to surpass or tie with closed-source SOTA models (Sora/Kling), which demonstrates the strong potential of TTT.

**Weaknesses:**

- Tab. 5 should include a comparison of the computational cost.
- Regarding the long-video baselines, the paper would be more sound if a more comprehensive set could be included [1,2]

[1] FIFO-Diffusion: Generating Infinite Videos from Text without Training

[2] Skyreels-v2: Infinite-length film generative model

- The paper lacks discussion and comparison with several recently accepted works on long-video generation.

[1] Zhao et al., Riflex: A Free Lunch for Length Extrapolation in Video Diffusion Transformers (ICML 2025).

[2] Tan et al., FreePCA: Integrating Consistency Information Across Long-Short Frames in Training-Free Long Video Generation via Principal Component Analysis (CVPR 2025).

[3] Lu et al., FreeLong: Training-Free Long Video Generation with SpectralBlend Temporal Attention (NeurIPS 2024).

[4] Cai et al., DitCtrl: Exploring Attention Control in Multi-Modal Diffusion Transformer for Tuning-Free Multi-Prompt Longer Video Generation (CVPR 2025).

**Questions:**

See the Weaknesses section.

---

> ### Author Response · Authors · 2025-12-02
> **Response to Reviewer UF3o**
>
> ### Q1: Tab. 5 should include a comparison of the computational cost. ###
> We will provide the cost in the final release. However, the change in cost is marginal (less than 1%) as we simply pass a subset of the frames into Qwen.
>
> &nbsp;
> ### Q2: Compare against SkyReels-V2 and FIFO-Diffusion. ###
> We have conducted a qualitative comparison between our MCTS framework, FIFO-Diffusion [1], and SkyReels-V2 [2], which can be viewed in this slidedeck: https://docs.google.com/presentation/d/1Agi0E049WxfRjtCEmpNyhvOzAWv_4g8yIqhGK6idSMA/edit?usp=sharing.
>
> Here is the quantitative comparison of the methods using Gemini-Pro 3 is the VLM judge:
> | Method | Temporal Consistency | Subject Consistency | Aesthetic Quality | Motion Smoothness |
> | :--- | :---: | :---: | :---: | :---: |
> | **FIFO** | 3.33 | 3.33 | 6.67 | 7.33 |
> | **MCTS** | 8.33 | 9.00 | 7.66 | 9.00 |
> | **SkyReels** | 8.33 | 8.66 | 7.66 | 9.33 |
>
> As demonstrated in the comparison, our MCTS framework achieves superior temporal/subject consistency and object permanence relative to both baselines. This performance advantage is attributable to our search mechanism and reward function, which explicitly optimize for these attributes over long horizons, rather than relying solely on the base model’s conditioning.
>
> Furthermore, we emphasize that our search-based approach is orthogonal to the techniques used in these baselines:
>
> + **FIFO-Diffusion** relies on diagonal/lookahead denoising and latent partitioning.
>
> + **SkyReels-V2** utilizes a Diffusion Forcing transformer architecture combined with a sliding window inference strategy.
>
> Our MCTS method functions as an inference-time optimization that can be applied on top of such autoregressive or diffusion-forcing processes by sampling new noise initializations to generate and evaluate a set of plausible continuations. Consequently, rather than being a competing alternative, SkyReels-V2 represents a powerful potential backbone for our MCTS framework, allowing for the combination of its strong generation capabilities with the long-term planning benefits of search.
>
>
> &nbsp;
> ### Q3: Discussion and comparison with several recently accepted works on long-video generation. ###
> While the suggested works—RIFLEX [1], FreePCA [2], FreeLong [3], and DiTCtrl [4]—represent significant advancements in long video generation, they fundamentally operate by modifying the underlying architecture of the base model. In contrast, our MCTS framework operates as an inference-time planning algorithm. Consequently, our approach is orthogonal to these methods and can be applied on top of them to further enhance long-term coherence.
>
> + **RIFLEX [1]** addresses the failure modes of positional embeddings (RoPE) by identifying and reducing an "intrinsic frequency" to prevent temporal repetition and motion deceleration. While effective for 2x or 3x extrapolation, it remains a deterministic extension of the base model's capabilities and does not actively search for optimal narrative continuations.
>
> + **FreePCA [2] and FreeLong [3]** focus on feature-level consistency. FreePCA utilizes Principal Component Analysis to decouple globally consistent appearance features from local motion intensity. Similarly, FreeLong employs SpectralBlend Temporal Attention to balance low-frequency global consistency with high-frequency local details. Both methods mitigate signal degradation but lack a mechanism to evaluate multiple potential futures or plan for long-term object permanence.
>
> + **DiTCtrl [4]** treats generation as a temporal editing task, utilizing Mask-Guided KV-Sharing and latent blending to ensure smooth transitions between prompts. While this ensures smoothness, it does not solve the planning problem of what should happen next to maintain logical coherence over long horizons.
>
> **Conclusion:** Long video generation is fundamentally a planning problem. Search is the primary mechanism for long-term planning. Our MCTS-based approach introduces a reward-guided search process that selects the most plausible continuations from a set of possibilities. This is distinct from the internal architectural modifications proposed in [1-4]. As such, our method is complementary; for instance, one could use RIFLEX or FreeLong as the backbone generator within our MCTS search node expansion, combining the signal-fidelity benefits of those methods with the long-term planning capabilities of search.

---

### Official Review · Reviewer_kWNU · 2025-11-01

**Soundness:** 2
**Presentation:** 3
**Contribution:** 3
**Rating:** 4
**Confidence:** 4

**Summary:**

The author introduces a Multi-Tree MCTS variant that improves exploration in continuous generation spaces.

**Strengths:**

1. The paper is well written.

2. The author introduces a Multi-Tree MCTS variant that improves exploration in continuous generation spaces. It is interesting.

**Weaknesses:**

1. I would like to know the time it takes to generate a 1-minute video with and without using your MCTS, and provide a quantitative comparison of the results.

2. The biggest issue with video generation is the excessive time consumption. This MCTS could make generating a long video take 24 hours, potentially requiring 20 times more time.

3. It is difficult to implement. The biggest challenge of this model is the accurate training of the Process Reward Model and Outcome Reward Model. As we know, video quality is hard to evaluate (the error rate of evaluation is high). Any slight error in the evaluation of these two models could lead to a massive search error.

4. MCTS does not have good robustness for the Process Reward Model and Outcome Reward Model.

5. I believe the author should focus on reinforcing the video model with reinforcement learning instead of using TTS, as it is a more efficient and practical solution.

**Questions:**

see weakness

---

> ### Author Response · Authors · 2025-12-02
> **Response to Reviewer kWNU (part 1/2)**
>
> ### Q1: Time to generate 1 minute video with and without MCTS ###
>
> In Table 2, we provide the time cost of generating 20 seconds of video. The baseline autoregressive method requires 0.5 GPU hours, while MCTS-guided generation requires 20 GPU hours.
>
> Extrapolating linearly, the overhead for generating a 1-minute video (60 seconds) requires 1.5 GPU hours using the autoregressive approach, and 60 GPU hours using MCTS. This linear extrapolation assumes each 20 second chunk is generated with a separate MCTS tree, conditioned on the previous segment.
>
> Although this seems like a stark difference, many practical techniques exist (Table 1) that can significantly reduce the time required to generate the videos on the order of 100 times - down to minutes. As the cost of generating a single video continues to drop along with improvements in the software/hardware stack, our approach becomes increasingly viable.
>
> &nbsp;
> ### Q2: MCTS could make generating a long video take 24 hours, potentially requiring 20 times more time ###
>
> **Table 1: Practical techniques to reduce generation time**
> | Optimization Technique | Performance Driver | Speedup Factor (Relative to Baseline 1.0x) | Notes/Mechanism |
> | :--- | :--- | :--- | :--- |
> | Baseline (Vanilla Attention) | Unoptimized Forward Pass | 1.0x | The original, unaccelerated performance of the model using all sampling steps. |
> | DMD Distillation | Sampling Steps Reduction | 15x ~ 50x | Achieved by reducing the required number of generative sampling steps (e.g., from 50 steps to 4 steps or 1 step). |
> | TensorRT-LLM | Inference Engine Optimization | ~2 to 3x | General acceleration by optimizing the LLM graph and runtime engine. |
> | FP4 Quantization | Low-Bit Precision | ~1.3x | Reduces memory bandwidth requirements by storing parameters in 4-bit precision. |
> | Hardware Improvement | B200 vs H100 | ~2 to 3x | The B200 GPU offers better performance than the H100 GPU for video generation models. |
>
> Applying these optimizations in parallel could yield upwards of a 600x reduction, bringing generation down from 20 GPU hours to a few GPU minutes. Moving forward, combining these optimizations can be a practical approach.
>
> We contend that the focus of this work is **algorithmic and conceptual contribution to the general field**, rather than immediate engineering practicality.
>
> **1) Conceptual Focus**: The primary contribution of this paper is to propose and demonstrate a new algorithmic direction: the benefit of applying **Monte Carlo Tree Search (MCTS) for Test Time Scaling** in long video generation.
>
> **2) Future Viability:** We assert that it is **unfair to assess the paper solely on the basis of current engineering practicality**. With ongoing improvements in distillation, quantization, inference optimization, and hardware acceleration, our MCTS-based approach—already viable today—will continue to gain efficiency and robustness in production environments.

---

> ### Author Response · Authors · 2025-12-02
> **Response to Reviewer kWNU (part 2/2)**
>
> ### Q3: It is difficult to implement. The Process Reward Model and Outcome Reward Model are the biggest challenge. Any slight error in the evaluation of these two models could lead to a massive search error. ###
>
> We acknowledge that the current visual reward model is imperfect. Nevertheless, our framework is capable of generating **high-quality and coherent long videos** (generated video can be found in this slidedeck: https://docs.google.com/presentation/d/1Agi0E049WxfRjtCEmpNyhvOzAWv_4g8yIqhGK6idSMA/edit?usp=sharing), which validates the importance of studying **Test Time Scaling** of video generation models—a common practice for LLMs but novel for diffusion models.
>
> This robustness is a direct consequence of the **exploration mechanism inherent in MCTS**. Rather than pushing on a single trajectory suggested by a noisy reward, MCTS actively explores around to maintain a well-balanced trade-off between exploration and exploitation. This capability allows us to generate state-of-the-art (SOTA) videos even with the existing reward model, proving that the underlying scaling framework is sound. However, this is not true for other exploitation-only search algorithms such as beam and greedy  search, which heavily rely on accurate rewards.
>
> Our framework is designed to be future-proof: **As reward models and visual evaluation methods improve, the performance of our MCTS framework will also improve**. It is imperative to keep improving the reward model to bridge the gap between current model and professional human performance. We leave this as a future work, and this should not undermine this paper’s merit in contributing a novel Test time scaling algorithm for video generation.
>
> &nbsp;
> ### Q4: MCTS does not have good robustness for the Process Reward Model and Outcome Reward Model ###
> We recognize the concern that reliance on imperfect reward models introduces noise into the search landscape. However, we contend that this perceived weakness highlights the **necessity and robustness** of MCTS for this problem.
>
> MCTS is uniquely suited for noisy, uncertain environments because it is designed to balance exploration with exploitation, enabling it to probe regions with lower expected value (under the noisy reward model) that may contain the true optimal solution. This inherent resilience makes MCTS the only viable search suite for this task, particularly when compared to a purely greedy approach, which would inevitably be trapped by local optima induced by the noisy signal.
>
> We stress that the limitation of the current reward model is an **external, orthogonal challenge** that affects all methods. We confirm that our framework's performance and robustness will scale directly with advancements in underlying models and verifiers, confirming its validity as a long-term algorithmic direction.
>
> &nbsp;
> ### Q5: Focus on reinforcing the video model with reinforcement learning instead of using TTS ###
> TTS has shown promising results in LLMs, and when applied to video generation can beat Sora-1 in terms of length and quality and the latest Kling model in terms of length, even using the Cosmos-Predict 2 backbone (https://docs.google.com/presentation/d/1Agi0E049WxfRjtCEmpNyhvOzAWv_4g8yIqhGK6idSMA/edit?usp=sharing). As TTS for video generation matures and the cost of video generation decreases, it will become more practical.
>
> We appreciate the suggestion to use a pure RL approach, but we argue that our focus on Test Time Scaling (TTS) introduces a necessary and orthogonal direction for scaling generative models.
>
> **RL Limitations:** RL-based approaches are inherently **sample-inefficient** and require a distinct policy to be learned and retrained for every underlying model backbone. Moreover, the policy would need to be fine-tuned for every base model for ideal performance.
>
> **TTS Advantage (Plug-and-Play)**: In contrast, our TTS method is designed to be model-agnostic and requires **no additional training**. This allows it to be immediately applied to powerful existing backbones, such as the Sora-1 and Kling models.
> We view RL as complementary; using it to strengthen the underlying model would only enhance the performance of our TTS method further.

---

### Meta-Review · Area_Chair_zvZi · 2026-01-05

**Summary:**

kWMU has concerns on the significantly increased computational cost (x20) of the proposed method over the baseline method, and challenges of the method in terms of sensitivity to the reward model. The reviewer didn’t respond to the rebuttal but authors acknowledged that the significant cost increase while arguing that other methods can help speed-up. UF3o rated marginal acceptance with suggestions on adding comparisons to more methods. The reviewer didn’t respond to the rebuttal, but from authors’ new experiments, it doesn’t seem the proposed method is clearly better than SkyReels while incurring significantly higher computational cost. Reviewer 6AqY raised a similar concerns as kWMU on the sensitivity to Rewards, and suggested sensitivity to parameters.

**Reviewer Concerns:**

Some of the clarification questions are addressed in the rebuttal.

The significantly increased computational cost seems acknowledged in the rebuttal, and not addressed well.

Also comparison to SkyReels seems not showing clear advantage in performance.

**Reviewer Scores:**

Reviewers would probably keep their original scores.

---

### Decision · Program_Chairs · 2026-01-26

Reject